# Outdoor Radon as a Tool to Estimate Radon Priority Areas—A Literature Overview

**DOI:** 10.3390/ijerph19020662

**Published:** 2022-01-07

**Authors:** Igor Čeliković, Gordana Pantelić, Ivana Vukanac, Jelena Krneta Nikolić, Miloš Živanović, Giorgia Cinelli, Valeria Gruber, Sebastian Baumann, Luis Santiago Quindos Poncela, Daniel Rabago

**Affiliations:** 1“VINČA” Institute of Nuclear Sciences—National Institute of the Republic of Serbia, University of Belgrade, 11000 Belgrade, Serbia; icelikovic@vin.bg.ac.rs (I.Č.); pantelic@vin.bg.ac.rs (G.P.); vukanac@vinca.rs (I.V.); jnikolic@vinca.rs (J.K.N.); milosz@vin.bg.ac.rs (M.Ž.); 2European Commission, Joint Research Centre (JRC), 21027 Ispra, Italy; 3Laboratory of Observations and Measurements for the Climate and the Environment, National Agency for New Technologies, Energy, and Sustainable Economic Development (ENEA), 90141 Palermo, Italy; 4Austrian Agency for Health and Food Safety, Department for Radon and Radioecology, 4020 Linz, Austria; valeria.gruber@ages.at (V.G.); sebastian.baumann@ages.at (S.B.); 5Radon Group, University of Cantabria, 39011 Santander, Spain; luis.quindos@unican.es (L.S.Q.P.); daniel.rabago@unican.es (D.R.)

**Keywords:** outdoor radon concentrations, literature overview, radiation risk, indoor radon concentrations, radon priority areas

## Abstract

Doses from the exposure to outdoor radon are typically an order of magnitude smaller than those from indoor radon, causing a greater interest on investigation of the latter for radiation protection issues. As a consequence, assessment of radon priority areas (RPA) is mainly based on indoor radon measurements. Outdoor radon measurements might be needed to guarantee a complete estimation of radiological risk and may help to improve the estimation of RPA. Therefore, authors have analysed the available literature on outdoor radon to give an overview of outdoor radon surveys and potential correlation with indoor radon and estimation of RPA. The review has shown that outdoor radon surveys were performed at much smaller scale compared to indoor radon. Only a few outdoor radon maps were produced, with a much smaller density, covering a larger area, and therefore putting doubt on the representativeness of this data. Due to a large variety of techniques used for outdoor radon measurements and requirement to have detectors with a high sensitivity and resistance to harsh environmental conditions, a standardised measurement protocol should be derived. This is no simple endeavour since there are more applications in different scientific disciplines for outdoor radon measurements compared to indoor radon.

## 1. Introduction

Radon is a naturally occurring, noble gas whose isotopes are all radioactive. The two most abundant isotopes ^222^Rn (called radon) and ^220^Rn (called thoron) are part of the natural decay chains of ^238^U and ^232^Th and originate directly from the decay of ^226^Ra and ^224^Ra isotopes, respectively. ^226^Ra is practically ubiquitous in rocks and soils and in any construction material made from them. Radon reaches the outdoor environment mainly by exhaling from the top soil. Radon can reach the atmosphere also from ground waters, natural gas, oceans and because of human’s activities. Due to its properties, atmospheric radon (outdoor radon), measured as radon concentration in air (Bq m^−3^), radon flux measured in radon exhaled per surface unit and per time unit (Bq m^−2^ s^−1^) and radon in soil gas (Bq m^−3^) are used in different scientific disciplines: (1) as a tracer gas for the estimation of greenhouse gases [1,2,3,4,5,6]; (2) to investigate mixing processes in the atmospheric boundary layer and the evolution of air pollution [7,8,9,10,11,12]; (3) as a tracer for dynamic processes such as: volcanic activity and earthquake prediction, caves air exchange, groundwater movements [13,14,15,16,17,18,19,20,21]; (4) as a tracer of Non-Aqueous Phase Liquids (NAPL) contamination [22]; (5) to estimate the health risk due to exposure to radon and its progeny [23,24].

The World Health Organization (WHO) has identified ^222^Rn (radon) as one of the leading causes of lung cancer after smoking. It is responsible for 3–14% of all deaths caused by lung cancer [23]. The European Commission included radon in the legal framework within the Basic Safety Standards (BSS) Directive—Directive 2013/59/Euratom [25]. According to this directive, EU member states are obliged to develop a Radon Action Plan to address long-term risks from radon in dwellings, buildings with public access and workplaces. Further, Article 103 requires Member States to identify areas where the radon concentration (as an annual average) in a significant number of buildings is expected to exceed the relevant national reference level, called radon priority areas (RPA). In such RPAs remedial actions in workplaces with indoor radon concentration above the reference level should be taken [25]. Similarly, for non-EU countries International Atomic Energy Agency (IAEA) BSS [26] requires radon surveys to be performed and provides guidelines how to conduct them [27].

The need for implementation of the EU-BSS has triggered several activities, mainly in Europe. Although many national surveys have been performed before the 2013 BSS, several radon surveys have started or are being repeated recently [28,29]. In line with the BSS directive and the obligation of member states to identify RPAs, a lot of efforts have been made to evaluate concepts and definitions of RPAs and to give description on how to delineate them [30,31]. The recently finished metrological research project: 16ENV10 “Metrology for radon monitoring” (MetroRADON—http://metroradon.eu (accessed on 21 November 2021)) aimed to develop calibration methods and measurement techniques to assist countries to establish their national radon action plans [32,33]. Reports are available at the EURAMET website [34]. Some of the objectives were to analyse and develop methodologies for the identifications of RPAs, to develop the concept of Radon Hazard Index and to investigate the relationship between indoor radon concentration and geogenic radon [35].

The designation of the RPA is mainly done based on indoor radon measurements or the geogenic radon potential. The latter could be estimated using different geogenic parameters such as radon in soil gas and soil permeability, radionuclides contents in soil and rocks, and also radon flux and outdoor radon investigated in the present paper. Within the MetroRADON project, mapping exercise was organized aiming to test applicability of different mapping methods, developed in various countries, on the data set in another country and to test to what extent different methods applied on the same set of data can give similar results [36]. In most of these proficient mapping methods, various types of input parameters were used: (1) measured quantities such as: indoor radon concentrations, radon in soil gas, permeability, radionuclide content in soil, terrestrial gamma dose rate; (2) geological data: type of soil, hydrology and faults; (3) meteorological data: precipitation, pressure; and (4) anthropogenic factors such as building characteristics, living habits (influencing ventilation rate) [36]. However, one can notice that the data on outdoor radon and radon flux were generally missing.

The possibility to use the radon flux and outdoor radon for a RPA designation and for climate research is investigated in a new metrological project, EMPIR 19ENV01 “Radon metrology for use in climate change observation and radiation protection at the environmental level” (traceRadon—http://traceradon-empir.eu (accessed on 21 November 2021)). This project aims to develop metrological capacity to measure low levels of radon in the environment, which can be used to determine emission strategies of greenhouse gases [37]. The project will go beyond the state of the art since it attempts to develop improved methods to identify RPA by including outdoor radon concentrations and radon flux measurements. This will extend the concept of Geogenic Radon Hazard Index (GRHI), implemented in the preceding EMPIR 16ENV10 MetroRADON project, that up to date has only static data, to include dynamic data [35].

In this article, first research results from the traceRadon project are presented. The authors aim to provide a literature overview of outdoor radon surveys mainly focussing on the correlation with indoor radon and estimation of RPA.

## 2. Outdoor Radon

In general, radon in outdoor air does not represent a significant health risk to the general population. The reason for this is twofold. Radon exhaling from the ground is quickly diluted to low concentrations in air. Typical outdoor radon concentration ranges between 1 and 100 Bq m^−3^, with an estimated annual average of around 10 Bq m^−3^ [38], while indoor radon concentrations range from a few Bq m^−3^ to more than 70 kBq m^−3^ [39] with an average of around 40 Bq m^−3^ [40]. In addition, it is estimated that people spend 20% of their time outdoors and consequently, the dose due to the exposure to average outdoor radon concentration is an order of magnitude smaller than the dose received from the exposure to average indoor radon concentration [40].

There are no specific regulations/recommendations regarding the outdoor radon concentration, and hence such measurements were not performed on as large scale as indoor radon measurements.

But the following arguments explain why knowing the outdoor radon concentrations is of interest for radiation protection issues:Knowing the outdoor radon concentration can serve as a baseline to assess anthropogenic contribution to radon exposure. Some locations have potentially hazardous levels of outdoor radon concentrations due to anthropogenic contribution like NORM repository such as phosphogypsum, uranium mining and milling, dumps from uranium mining waste, thermal power plants or from any mining industry that has material with concentrated ^226^Ra as a waste, or material that is being exposed to the accessible environment [41,42,43,44].Outdoor radon could contribute to radon dose especially in regions where outdoor radon concentrations are higher or close to the indoor ones [45]. In addition, there are regions with high level of natural radiation that can have much higher annual average outdoor radon concentration than the worldwide average. For example, in high natural radiation environment of Niška Banja, Serbia (a radon spa) average annual outdoor radon concentration was 57 Bq m^−3^ with a maximal value of 168 Bq m^−3^ [46], while in Norway, a maximal radon concentration of 114 Bq m^−3^ was measured in Ullensvang commune [47,48].Indoor radon concentration over a large region is usually described by a lognormal distribution [49,50,51,52]. From the parameters of the lognormal distribution such as geometric mean (GM) and geometric standard deviation (GSD), a percentage of dwellings above some reference level (RL) can be derived, which could be used to estimate whether some region is an RPA. Significant deviations from the lognormal distribution, observed in certain cases could be attributed to geological variability of an area, leading to a mixture of different lognormal distributions [53,54]. It was shown that disregarding the outdoor radon concentration in data analysis of indoor radon surveys will cause a deviation from a lognormal distribution. The higher the outdoor value, the higher would be the underestimation of the GSD. Figure 1 shows frequency distributions of indoor radon concentrations, uncorrected and corrected for outdoor radon concentrations [55,56]. After subtracting the outdoor concentrations from the indoor radon data, a distribution closer to a lognormal distribution is obtained. Omitting to correct for outdoor radon concentrations will have as a consequence an underestimation of percentage of dwellings exceeding a reference level and could have an implication in classification of RPAs [57].The sensitivity analyses have shown that failing to correctly assess radon exposure reduces the ability of epidemiological studies on small sample size to detect an effect due to radon [58,59].And finally, outdoor radon, together with the radon flux will be used as input variables to improve GRHI, which is one of the approaches used for the designation of RPA [35]. Deriving methods for identification of RPA by including outdoor radon concentrations will be one of the goals of ongoing EMPIR 19ENV01 traceRadon project [37].


## 3. Properties of Radon in Outdoor Environment

Radon concentration in soil gas has at least 3–4 orders of magnitude higher concentrations than the radon in the open air. Therefore, there is a strong gradient between soil and outdoor air. This gradient in concentration between soil gas and outdoor radon is continuously maintained due to continuous generation of radon atoms from the long-lived mother nuclei and decay of radon in atmosphere. The amount of radon concentration in the open environment depends on the source, exhalation rate (amount of activity released per unit time through a unit surface) and dilution processes in the open environment. In other words, radon concentrations in the environment depend on 1) the content of radon precursors (^238^U and ^226^Ra) in the ground/rocks—responsible for production of radon in soil, 2) on properties of ground/rocks such as: size of mineral grains, porosity, permeability, moisture content—responsible for radon emanation from the soil and its transport toward the troposphere, and 3) the type of soil cover—that influence the radon exhalation rate [60].

There are numerous factors, mainly meteorological conditions, which can influence radon flux from soil and consequently outdoor radon concentration. An increase of atmospheric pressure, rainfall and snowfall decreases the radon flux, while an increase of wind speed or temperature increases it [61].

Since many parameters influence outdoor radon concentrations, it exhibits different type of variations which are briefly described in following subchapters. Diurnal variations are discussed in Section 3.1, seasonal variations in Section 3.2. Variation of annual radon concentrations are discussed in Section 3.3. Radon concentrations at different heights from the ground, but restraining to heights relevant to population exposure are discussed in Section 3.4. And finally, spatial variability of outdoor radon concentrations is mentioned in Section 3.5.

### 3.1. Diurnal Variations of Outdoor Radon Concentrations

Once reaching the atmosphere, radon is transported via turbulent air mixing characterised by eddy diffusivity. It depends on the vertical variations of wind velocity and is limited by the radioactive decay. Assuming no removal of radon and steady state conditions, Jacobi and André have estimated concentration of radon and its progeny as a function of height from the ground for four typical mixing scenarios in the troposphere (strong, normal, rather weak and strong inversion) [62]. Eddy diffusivity is described by the turbulent diffusion coefficient. The model predicts a strong dependence of radon concentration on atmosphere mixing. For the same height from the ground, assuming constant exhalation rates, minimal values were predicted for strong mixing and almost 100 fold higher values for a strong inverse mixing. These differences are more pronounced at near ground level than at higher altitudes [62]. References supporting the derived model and a few available measurements of eddy diffusivity are given in the review article of Porstendörfer [61].

Diurnal variations of radon concentrations can be explained by diurnal variations of mixing in the lower atmosphere. Temperature gradient between 1 m and 5 m height from the ground is periodically positive and negative. During the night and in the early morning, there is a positive temperature gradient causing an inverse atmosphere mixing, which leads to the highest outdoor radon concentrations. On the other hand, at noon and during the afternoon there is the largest negative temperature gradient causing the strongest atmosphere mixing, causing the lowest radon concentrations [63,64]. Diurnal variation of the radon concentration has been a subject of many investigations [63,64,65,66,67,68,69,70,71,72] and typical behaviour is presented in Figure 2 (taken from [66]).

This 24 h cycle is more pronounced in the summer during the cloudless days with light winds and large diurnal temperature range, compared to cloudy weather with a lower temperature gradient during the day and the night, the latter had small or no variations of outdoor radon concentrations [61,66]. A deviation from daily pattern is observed also in days with a strong wind and precipitation [65].

Aquilina and Fenech have observed statistically significant negative correlation between outdoor radon concentration and wind speed in two regions in Malta, with R values of −0.74 and −0.88 and beta coefficients of −0.53 and 0.68. The data indicate that an increase of wind speed by one unit will decrease outdoor radon concentration by roughly one half and one third, respectively [73].

In L’Aquila, Italy, outdoor radon measurements were performed continuously for 2 years with 5 min sampling time, while meteorological parameters were measured with a resolution of 5 s. Both data are averaged hourly. Typical diurnal cycle was observed. With a stable atmosphere, mostly during the night radon accumulates at the surface, while with sunrise, turbulent vertical mixing starts decreasing radon concentration at the surface. The effect of wind is anticorrelated with radon with a correlation factors between −0.49 and −0.61 for different seasons. Temperature shows expected negative correlation ranging from −0.43 to −0.72, while relative pressure exhibits positive correlation with radon ranging from 0.25 to 0.58 [72].

Continuous measurements of outdoor radon concentrations and meteorological parameters were performed in a typical rural site in South India. Data analysis have revealed a high negative correlation parameter of −0.75 between outdoor radon concentration and temperature. A statistically significant positive correlation with R value of +0.66 was observed between outdoor radon and humidity [74].

Similar negative correlation of R = 0.73 was observed between outdoor radon progeny concentration and temperature. Measurements were performed daily in 3 months of winter period [75].

Diurnal variations of radon concentration depend also on geological location of a measuring site. For example, in river valleys there are typically strong night inversions that will cause large variation of radon concentration, while at the nearby hillside area, night inversions are much less pronounced, leading to a smaller variation of outdoor radon [76], which will be discussed more in Section 3.5.

### 3.2. Seasonal Variation of Outdoor Radon Concentrations

Similarly, seasonal variation of radon behaviour can be observed.

A study has been performed monitoring continuously indoor and outdoor radon concentrations in one location in Chiba, Japan, for 4 years using the active device AlphaGUARD [68]. It was found that diurnal changes are more pronounced for outdoor radon than those of indoor radon [68]. Furthermore, monthly gradients are more pronounced for outdoor radon than for indoor radon. The maximum observed outdoor radon concentration was in winter, while for indoor radon concentration in summer. In this study the data indicate that indoor and outdoor values are negatively correlated with Pearson correlation factor of -0.18 [68]. Differently, a linear relation was found between indoor and outdoor equilibrium equivalent concentrations of radon-222 with correlation coefficient of R = 0.9 [75]. Other results about indoor and outdoor correlation are discussed in paragraph 6.

Usually, indoor radon concentrations are highest in winter, during the heating season. Another 4 years continuous measurement of outdoor radon using a scintillator was performed in Milan, Italy. A similar seasonal pattern was observed: higher concentrations in winter (around 15 Bq m^−3^) than in summer (around 5 Bq m^−3^), by the factor of 3, as could be seen in Figure 3 taken from Sesana et al. [66].

Higher values in winter were also measured in Turkey (by a factor of 2) [77], in a nationwide survey in Japan [78] and China (by an average factor of 1.8) [79]. However, other references indicate that the lowest values were in winter, presumably due to a snow cover, large humidity or strong winds [46,80,81,82]. Finally, in the measurement campaign performed in Slovenia, no significant difference between outdoor radon concentrations in any of the 4 seasons was found [45].

### 3.3. Variation of Annual Outdoor Radon Concentrations

It is also interesting to investigate annual variations of outdoor radon concentrations. The magnitude of the variations can help to estimate how representative the annual average outdoor concentration is, obtained at the same location in different years. It was found that the annual average outdoor radon concentration measured for a few consecutive years did not show significant (<15%) differences between annual concentrations [83]. Another experiment that was performed in 4 consecutive years has shown that variations of annual average radon concentration were up to 15% and up to 30% for 2 locations, respectively [84]. In 4 years of continuous monitoring in Milan, Italy, variation of annual radon concentrations were up to 13%, agreeing with previous results [66]. No statistical differences in 3 consecutive years were observed in nationwide survey in Germany also [85].

In the period of 3 years, continuous measurement of outdoor radon was performed in Urbino, Italy, using AlphaGUARD located 3 m above the ground. Data were correlated with temperature, pressure and humidity and a good correlation (0.62) between difference in maximal and minimal radon concentration and maximal temperature difference in one point was observed. It was underlined that variations of radon concentrations could be used to separate variations of pollutant concentrations that are due to variation in the emission fluxes and variations in meteorological conditions [86].

Continuous outdoor radon measurements have been performed in Bratislava (Slovakia) during the years 1991–2009 and the results show that the average monthly radon activity concentrations present seasonal variations; however, the individual years differ from each other quite considerably with the values of the average annual radon activity concentrations varied from 4.1 to 7.2 Bq m^−3^ [87].

### 3.4. Radon Concentrations at Different Heights from the Ground

As soil gas radon concentration is of the order of 10 kBq m^−3^ to 100 kBq m^−3^, while worldwide average outdoor radon concentration is around 10 Bq m^−3^, a strong gradient of radon concentration at low altitude would be expected. Doi and Kobayashi have investigated a vertical distribution of radon concentration by placing discriminative radon/thoron detectors in duplicate at 10 different locations and at 5 different heights from the ground: 0.04 m, 0.15 m, 0.25 m, 0.7 m and 1.0 m. Detectors were deployed for 3 months in order to investigate seasonal variations as well. The values of annual average radon concentration for each location did not show any significant difference. Obtained annual average was 3.85 ± 0.19 Bq m^−3^ [88]. On the other hand, thoron concentrations show a strong decrease with the altitude, which is a consequence of its short half-life (55.8 s). The lack of a concentration gradient between different heights is interpreted as the interplay between vertical turbulent diffusion responsible for the radon transport from the soil toward lower layers of the troposphere and atmospheric mixing responsible for an exchange of air masses over large regions [88]. A similar experiment with detectors placed at 0.25 m, 0.5 m, 1.0 m, 1.5 m and 2.0 m, was performed at four different locations and no evidence for dependence on the height was observed [84]. Also, in order to investigate whether sampling is representative of the air that population breath, detectors were placed at 1 m, 1.5 m and 2 m in one experiment and between 1.5 m and 3 m in another [77,89]. No significant differences were observed between radon concentrations measured with these detectors.

For the sake of completeness, it should be mentioned that the variation of radon is measured also at greater heights, for example at 10 m and at 100 m above the ground [10]. This kind of investigation is interesting for the investigation of mixing processes in atmospheric boundary layer and will not be discussed in this review, but readers are encouraged to check [10] and reference therein.

### 3.5. Spatial Variability of Outdoor Radon Concentration

It was already mentioned few times that radon is sensitive to temporal changes of meteorological parameters and therefore is used as an indicator of vertical exchanges in the atmosphere [64]. Orography (i.e., geomorphology) of the terrain affects the local and regional weather conditions such as temperature, cloudiness, solar radiation, wind speed and wind direction [90]. In Slovakia, outdoor radon concentrations were continuously measured at four locations having different orography from the flatland to hilly terrain. At the measurement time, similar meteorological conditions were at all four locations, leading to high correlation between measured outdoor radon concentrations at the reference site [87] and other three locations (R = 82, R = 64 and R = 55) [90].

The diurnal variation at the reference site ^87^and hilly location exhibit a notable shift of up to 5 h in daily maximum. Namely, solar radiation was shorter due to shielding from the nearby hills, which tend to earlier decrease the vertical mixing. The effect is more pronounced in summer when influence of the solar radiation is more prominent [90].

Results indicate that if radon data ought to be representative for a larger area, orography of the locality, as well as type of soil should be carefully chosen [90].

It was shown also that radon time series are different for different types of communities: urban, suburban and rural [65].

In general, outdoor radon disperses fast from its source. However, it was shown that specific topology, i.e., configuration of terrain could prevent radon from fast dilution and consequently could be measured farther from its source [41]. Similarly, the highest values in outdoor radon survey performed in Japan were observed in a basin surrounded by mountains, so it is speculated that the convection of the air might be hampered by the topographical effects [78]. The effect of the location on diurnal variability is already mentioned in previous chapter observing a strong inversion in river valley during night and less pronounced variation at the hill side [76].

Radon flux from the ocean is several orders of magnitude smaller than the flux from the soil and thus radon concentration above oceans are of the order of tens of mBq m^−3^, while inland, outdoor radon concentration typically varies from 1 to 100 Bq m^−3^. Therefore, radon concentrations measured over land are the lowest in coastal areas where mixing between low radon marine air and continental masses occurs [73,91]. In Ireland, a statistically significant difference between coastal and inland radon concentrations was found due to differences between radon levels over land and over seawater [92].

Similar behaviour was found in the national survey of Germany. It was noticed that the lowest radon concentrations were in the North in the coastal region and were increasing toward high mountain area in the south. Besides the effect of influence of low radon air masses from the sea, bedrock in the southern part of Germany has higher ^226^Ra concentrations. [85]. The higher the ^226^Ra content in bedrock/soil, the higher the outdoor radon concentration would be. Statistically significant difference was found between the average annual radon concentration measured above a travertine (being rich in ^226^Ra) and above the nearby alluvial terrain. Higher outdoor gamma dose rate was also indicator of higher outdoor radon concentration [46]. On the other hand, in Slovenia, elevated indoor radon concentrations were not found neither in the granite region with 2–3 times higher gamma dose rate than elsewhere in the country, nor around thermal and mineral waters rich in ^226^Ra (Slovenia). The reasons could be lower emanation from compact granite and clay, because these topsoil types act as a good barrier for radon transport [45].

Although there are a few main processes that are responsible for the amount of radon concentration in air, there are many parameters influencing radon behaviour. This was also nicely demonstrated by Fourier analysis of indoor and outdoor time series, which has shown that there is more noise in the time series from outdoor than from indoor radon measurements [68].

## 4. Measurement Methods

There are numerous methods for outdoor radon measurements. Radon and some of its progeny: ^218^Po, ^214^Po and ^210^Po are alpha emitters, while ^214^Pb, ^214^Bi, ^210^Pb, and ^210^Bi are beta emitters, whose decay is followed by a gamma emission. Therefore, depending on whether radon is measured directly, or indirectly by measuring its progeny with or without radon itself, any nuclear technique that measures alpha, beta or gamma radioactivity can be applied.

There are two typical classifications of measuring methods:depending whether measurement requires usage of electricity and/or an air pump for collection of samples or not, measurement can be active or passive;depending on the type of sampling, and/or measurement duration, measurement can be *instantaneous* in which the air is sampled at the order of minutes (grab sampling), *continuous* in which radon is monitored in real time, with a sampling time in a predefined interval of the order of minutes or hours; and *integrating* in which the concentration is integrated (averaged) over the whole measuring period (of the order of days, weeks or months).

The choice of measurement method depends on the purpose of the measurement. For investigation of fast processes like diurnal variation of radon concentration, or its correlation with meteorological parameters, an active device in continuous mode should be used. If the research goals are investigations of seasonal variations of outdoor radon or measurement of annual average outdoor radon concentration, a long-term measurement with a passive device would be a proper choice.

Measurement of outdoor radon concentration is more challenging compared to indoor radon measurements. Outdoor radon concentrations are on average one to two (but could be higher) orders of magnitude lower than indoor radon concentrations, requiring therefore monitors with a low background. In addition, detectors deployed outdoors are more exposed to meteorological conditions (precipitation, temperature, wind, exposure to sun).

Since the aim of this review article is to analyse literature on outdoor radon concentrations with regard to radiation health risk, it is not surprising that the majority of used methods within the investigated surveys were integrating measurements. Among passive detectors, solid state nuclear track detectors, polyallyl diglycol carbonate (PADC)—CR-39 were used in 40.7% of analysed surveys [45,46,77,82,83,89,92,93,94,95,96], polycarbonate foil of Makrofol in 7.4% [78,85], one survey (3.7%) with electrostatic integrating monitor based on LR115 [79]. One survey (3.7%) was performed by electret ion chamber (EPERM^®^) [81]. Active devices were used in 44.4% of the surveys, out of which: Two (7.4%) ionization chambers (AlphaGUARD) [74,97], four (14.8%) EEC Rn devices [75,98,99,100], five (18.5%) scintillators [66,87,91,101,102] and one (3.7%) Passivated Implanted Planar Silicon (PIPS) detector for alpha spectrometry [73]. Details on used methods are given in Table 1.

Measuring devices for outdoor radon used as a tracer for Green House Gases (GHG) or for investigation of mixing processes in atmospheric boundary layer are constructed with much higher sensitivity. These devices are constructed using 3 different measurement principles: one filter, two filters and electrostatic deposition. Intercomparison of these devices can be found in Grossi et al., [103] and references therein.

An example of continual monitoring of ^222^Rn in atmosphere has been carried out in Slovakia where a large volume scintillation chamber with a volume of 4.5 litres was used for the measurements [87]. The air for the analysis was sucked at a height of 1.5 m above the ground surface. The sensitivity of the scintillation chamber is 0.3 cpm at 1 Bq m^−3^ of ^222^Rn activity concentration in the entering air. Radon activity concentrations were recorded in two-hour intervals (12 data per day) [87].

ISO 11665-1:2012 standard provides a guidance for measuring ^222^Rn activity concentration in the air in order to assure representativeness of the sampled air. According to the standard, measurement shall be performed between 1 m and 2 m height above the ground, with no obstacles in, at least, 1 m diameter from the sampling location and within the inverted cone with a 140° opening above the sampling space (not including weather shelter) [104].

In none of the reviewed articles, was an explicit statement that the measurement of radon in air was performed according to the ISO 11665-1:2012 found. However, in almost all surveys detectors deployed between 1 m and 2 m, the exceptions are one survey in which detectors were deployed at 2.5 m [66], one deployed in the range from 1.5 m to 3 m [89], and a survey in which the information on height of measuring locations was not given [77].

In several surveys, special attention was paid to the reduction of the background of detectors, increase of the measurement precision and sensitivity. In the national outdoor survey of Slovenia, limitations of track etch detectors for outdoor measurements were underlined. The detectors were unavoidably exposed for at least three months to substantial changes of air temperature, humidity and precipitation, both rain and snow. Special attention was paid to detector manipulation such as storing and mailing (e.g., keep them indoors as short a time as possible) [45]. The usage of E-PERM was also discussed. It is low cost and convenient for long-term measurements. However, results need corrections for background radiation and the surface should be free of dust, which could be a problem for outdoor measurements [81].

In order to increase precision of long-term measurements by CR-39 detectors, a group of authors developed a special measurement protocol. It included pre-etching of detectors before its exposure in order to distinguish radon tracks from the background [92]. Wasikiewicz et al. have applied special treatment of detectors. Detectors have been housed in specially designed containers to counteract the effects of outdoor environment (like humidity, temperature). They were sealed in polyethylene bags becoming almost independent of relative humidity [95]. In order to avoid the change of physical properties when exposed to heat [105], after putting detectors in polyethylene bag, it was placed inside a plastic pot wrapped with aluminium to reduce effect of a direct sunlight. Due to measurements in a low radon environment, selection of detectors with the lowest possible background and highest sensitivity was made. Each pot included 2 detectors for 3 months exposure and one detector for one-year exposure. Correction for ageing effect was also taken into account [95].

In the German nationwide outdoor radon survey, detectors based on polycarbonate film (Makrofol^®^) were carefully designed to counterpart meteorological conditions. Polycarbonate film of 0.3 mm thickness was placed in an open diffusion chamber with a glass fibre filter and volume of around 360 cm^3^. Plastic cover was used to protect the chamber from meteorological and mechanical influences. Detectors were placed at 1.5 m above the ground to measure radon concentration in breathing air. Two detectors per location were deployed in order to reduce measurement uncertainty [85,106].

## 5. Outdoor Radon Surveys

Health hazard related to radon issue was first observed among miners and one of the first systematic studies were related to miners and mining sites [107,108]. Similarly, measurements of outdoor radon concentrations were first limited to uranium mining and milling sites, and afterwards outdoor measurements in residential areas were performed [45,109,110].

Anthropogenic contributions like mining industry, dumps from uranium mining, repositories of fly-ash and slag from thermal power plants or repositories of phosphogypsum could lead to potentially high outdoor radon levels. In a German survey dedicated to investigating the influence of mining and milling activities on outdoor radon, concentrations were found to range from 5 Bq m^−3^ to 1700 Bq m^−3^, mostly exceeding 50 Bq m^−3^ [106]. Many authors have reported that enhanced radon concentrations in the vicinity of uranium tailings pile, could not be detected at distances larger than 1 km. However, in some cases, topological and climatological characteristics of the area can prevent radon from its immediate dispersion [41].

In the area of Nowa Ruda, former coal mining area in Poland, mean radon concentration from 70 monitoring locations was 70 Bq m^−3^, with a maximum value of 131 Bq m^−3^ [111]. In former coal mine in Slovenia, similar daily average value of 80 Bq m^−3^ was found [112].

A review of influence of phosphogypsum, a by-product from the phosphate industry, on the environment was given by Rutherford and collaborators [113]. Gutiérrez-Álvarez and collaborators have shown that phosphogypsum repository influence outdoor radon concentration in nearby Huelva city, Spain [44].

Since anthropogenic contribution could cause hazardous levels of outdoor radon concentrations, outdoor radon concentration should be monitored or at least determined in industrial complexes that generate a high level of radioactive material/waste rich in radium and uranium content. This is especially true for complexes close to settlements or situated in regions with a specific terrain configuration that prevent immediate dispersion of radon.

Outdoor radon surveys were performed at a much smaller scale compared to indoor radon surveys. While national indoor radon surveys take range from 85 measurement locations in Malta to more than 500,000 locations in UK [28] maximal number of outdoor locations was 696 in national survey of Japan [78].

In Table 1, an overview of different outdoor radon surveys is given. Table includes:types of survey: National, regional, local and nationwide (across the country, but not national);number of measuring locations andbasic measurement protocol: method used, position height of the sampling and sampling time.

According to the reviewed literature, only 5 national surveys have been identified worldwide. Four of them are in Europe: Germany, Spain, Ireland and Slovenia and one in Japan in Asia. Two surveys covered states in the USA: Missouri and Iowa. There is a clear disproportion in the number and size of surveys between indoor and outdoor radon surveys [28]. Furthermore, the representativeness of the surveys was not discussed in the papers. The most common quality control test was duplicate detectors [45,68,78,82,83,89,94,95]. Blank tests were applied in China and USA [89,94].

Two outdoor radon surveys covering federal states of USA (Missouri and Iowa) and one covering a region in the state of Minnesota were performed in order to provide properly estimated annual effective dose necessary for epidemiological studies that link lung cancer risks with the exposure to radon [83,94]. Indoor and outdoor radon survey carried out in the home of one hundred lung cancer patients in Rize province in Turkey aimed to investigate the relationship between lung cancer patients and exposure to radon [77]. In Slovenia, the first outdoor measurements were limited to uranium- and mercury-rich sites. Afterwards, a national survey was performed to get information on radon background levels [45]. The aim of radon survey performed in China was to more precisely evaluate lung doses. The Survey was performed in 9 large cities in China [79]. Next nationwide survey, covering about 20% of population in 33 cities across China, was performed in order to investigate whether there was an increase in outdoor radon concentrations measured 20 years ago due to rapid industrial development [89]. Results revealed no increase of outdoor radon concentration. A national radon survey in UK found an average outdoor radon concentration of 4 Bq m^−3^ [114], however, authors did not have access to the publication so more detailed analysis is missing. In another survey performed at 69 locations in England, specially designed passive nuclear track detectors for measuring low outdoor radon concentrations were used [95]. The aim of research performed by [82] was to measure radioactivity levels in Fen Complex, Norway, a former Fe and Nb mining cite. In Lebanon, the investigation of indoor and outdoor Rn was performed due to a very large number of lung cancer cases and smoking rates [81]. In Ireland, first estimation of outdoor radon was from the national indoor radon survey, assuming that the difference between indoor and outdoor radon should follow a lognormal distribution [115]. In more recent survey of outdoor radon measurements in Ireland, the main objective of the survey was to perform pre-treatment of detectors in order to optimize them for low radon concentrations [92]. In the German national survey the aim was to determine natural radiation exposure due to outdoor radon and its progeny [85]. In Malta, after national indoor radon survey was performed, Aquilina and Fenech have investigated the influence of meteorological parameters on indoor and outdoor radon concentrations [73]. Regional radon survey in Syria was performed in order to estimate to what extent radon in soil gas, radon exhalation from soil, outdoor radon and activity concentration of ^226^Ra in soil contribute to indoor radon concentrations [102].

Outdoor radon concentrations in Montenegro were estimated theoretically. Each value of measured indoor radon concentration was reduced by a certain small value. Such derived values of indoor radon concentration were tested for distribution and the value whose correction of indoor radon concentration gave the best fit to lognormal distribution is considered to be outdoor radon concentration. The obtained value for outdoor radon concentration was 7 Bq m^−3^ [93].

Several surveys from different countries: Netherlands [99], Brazil [98], Russia [100] have measured radon progeny. The estimation of radon concentration from radon progeny concentration requires assumption of equilibrium factor, that was found to vary from 0.13 to 0.91 [116,117,118] and could introduce a large uncertainty. Another source of uncertainty could be different weather conditions (like precipitation causing fallout of radionuclides). Although calibration could reduce uncertainties, surveys on radon progeny will not be further discussed in this literature overview.

Descriptive statistics of outdoor radon surveys is given in Table 2. A range of measured concentrations is given, together with the geometric and arithmetic mean and geometric standard deviation.

In the case of the Spanish survey, there is a network of 25 automatic stations spread all over the Spanish country. Their locations are highly variable, ranging from rooftop stations in cities, to single stations on coastal borders or mountain ridges. The outdoor radon concentration is estimated by the pseudo-coincidences decay of ^214^Po/^214^Bi assuming equal concentrations of all the short lived radon progeny and constants during each measurement integration time. The monitoring system pumps in outdoor air at approximately 5 m^3^ h^−1^ through a moving glass–fibre filter where aerosol particles are retained which is associated to a ZnS plastic scintillator ^101^.

Outdoor radon maps covering the whole country were created in Germany with the resolution of 1 measuring location per 50 × 50 km^2^ grid cell [85], 1 detector per 20 × 20 km^2^ grid cell in Slovenia [45] and 3 maps in USA in Missouri, Iowa and Minnesota with around 40 km distance between measuring locations [83,94]. On the other hand, the European Indoor Radon Map is produced by averaging indoor radon concentrations within 10 km × 10 km grid cells [119] with some much smaller grids of indoor radon maps: 1 km × 1 km grid cells in United Kingdom [120,121,122] or even 0.5 km × 0.5 km [93].

Derived maps of outdoor radon are based on a small number of locations, over a large grid cell. It is not clear whether allocated values for outdoor radon concentrations in a certain cell could be considered as representative or not, especially having in mind that outdoor radon concentrations are influenced by so many factors, primarily the following: type of soil/bedrock, type of topsoil, topology of terrain and weather conditions. However, it would be difficult to obtain representative sampling with the similar density of deployed detectors as for indoor radon surveys, since it is difficult to provide a safe space to place detectors, especially in urban areas.

## 6. Correlation of Outdoor Radon with Other Measured (Radioactive) Quantities

Correlation of outdoor radon with indoor air, radon in soil gas, radon exhalation rate, activity concentration of ^226^Ra in soil and gamma dose rate are shown in Table 3.

One-year continuous measurements of indoor and outdoor radon concentrations were performed at one location in Bratislava, Slovak Republic. Analysis revealed high correlation between courses of indoor and outdoor radon correlations both during the day and during the year going up to R = 93 for daily courses in July ^123^. In Rize province, Turkey, besides the indoor and outdoor radon concentrations measured at around 100 locations, soil samples were taken from 80 locations. A weak correlation was found between indoor and outdoor radon concentration and activity concentration of ^226^Ra in soil. It is speculated that this could be due to other influences as well: soil porosity, pressure difference between indoor and outdoor and usage of buildings [77]. A good correlation (0.59) between outdoor radon concentration and radon in soil gas was obtained in Slovenia. In addition, it is mentioned that indoor and outdoor radon concentrations coincide well in a region of elevated radon concentrations, however giving only descriptive analysis [45,124]. In survey performed in China, a high correlation of 0.88 between ^226^Ra in soil [125] and outdoor radon concentration was observed [79], although data from some cities were not included in the analysis. In a radon spa in Serbia, although outdoor radon and gamma dose rate were not compared directly, it was found that outdoor radon concentration and gamma dose rate were higher above travertine (known for high ^226^Ra content) by the factor of 1.6 and 1.7, respectively, than above alluvium sediments [53]. Radon, thoron and gamma dose rate measurements were performed in Fen Complex in Norway, at a NORM site known for exceptionally high ^232^Th content in soil, up to 15500 Bq kg^−1^ [82], radon, thoron and gamma dose rate measurements were performed. The obtained thoron concentrations were up to 4996 Bq m^−3^. Statistical analysis has shown a moderate correlation between thoron concentration and gamma dose rate (0.56), as well as between radon and gamma dose rate (0.64) [82]. In Punjab, India, indoor and outdoor equilibrium equivalent radon concentrations EECRn were measured daily for 3 months in winter [75] In Lebanon, radon detectors of indoor and outdoor radon concentrations were simultaneously deployed [81]. Although the paper did not explicitly give correlation between indoor and outdoor radon concentrations, authors of this review have calculated correlation from the available data. The obtained value is 0.10, for the whole season, while the highest correlation was found to be 0.26 for autumn season. Since the standard deviation for outdoor radon concentrations is larger by 23% than its indoor counterpart, authors have suggested that weather-related patterns are contributing to the large variability in outdoor radon concentrations. By applying different ANOVA tests it was concluded that the location affects outdoor concentration, proximity of faults did not affect indoor or outdoor radon concentration and outdoor concentrations in summer are significantly higher than outdoor concentrations in winter [81,82]. No correlations were investigated in Ireland survey. However, it was found that radon concentrations inland are systematically higher than in coastal regions [92]. In Japan, a correlation factor of 0.62 was observed between outdoor radon concentrations and environmental gamma dose rate. However, it should be noted that outdoor radon and gamma dose rate measurements were performed at different locations [78] On the other hand, one year continuous measurements of outdoor radon concentration and gamma dose rate revealed no correlation between them (r = −0.013), which was explained by the stable gamma dose during the whole year, while outdoor radon concentration exhibited typical periodical behaviour ^74^In a German survey, a high correlation factor (0.93) between radon and equilibrium equivalent radon concentrations was obtained. A map of outdoor radon has shown low radon in the North due to the influence of low radon air masses from the Atlantic, and higher radon concentration in the southern part, due to bedrock with higher ^226^Ra concentrations [85].

In 20 measuring sites located in Japan, China and South Korea, radon flux, outdoor radon concentration and ^226^Ra in soil were measured. While weak correlation between outdoor radon concentration and activity concentration of ^226^Ra in soil was found, a significant correlation (0.79) was observed between outdoor radon and radon flux [96].

Good correlations were found only between outdoor radon concentration and radon flux in 2 out of 4 analysed governorates in Syria. Based on data from Table 1, in reference [102], authors of this review have determined the correlation between radon in air and radon flux, radon in soil gas, indoor radon and activity concentration of ^226^Ra in soil. Obtained correlations are given in Table 3.

In summary, correlation factors between outdoor and indoor radon concentration range from 0.1 to 0.93; between outdoor radon and radon in soil gas: 0.16–0.88; between outdoor radon and exhalation rate: 0.12–0.79; and between outdoor radon and gamma dose rate: −0.13–0.64. A wide range of correlation factors is not surprising since there are many parameters influencing measured quantities, and therefore, in order to estimate the influence of outdoor radon concentration on indoor radon concentration, a larger statistics with other parameters is required.

## 7. Conclusions

Although outdoor radon does not represent a significant health risk, its concentrations should be determined in order to obtain an accurate dose assessment, which is important for epidemiological studies. In addition, in the case of national or regional studies, it was shown that knowing outdoor radon concentrations is important for the correction of indoor radon values, which is further important for accurate parameterisation of indoor radon distribution. Not including outdoor radon concentration could lead to inaccurate estimation of percentage of houses above the reference level, which could further lead to inaccurate delineation of radon priority areas.

Measurement of outdoor radon concentration represents a challenge compared to indoor radon measurements, since concentrations outdoors could be a few orders of magnitude lower than indoor radon concentrations. Furthermore, detectors are exposed to harsh environmental conditions (precipitation, humidity, sunlight, wind) that can influence detector’s performances. A literature overview of outdoor radon surveys has revealed a usage of various types of detectors, deployed at different heights from the ground and with different types of shelters if any. For the purpose of harmonising and comparing data, a standardised measurement protocol should be derived.

In order to develop an optimised protocol for outdoor measurements, measurement techniques that are routinely used for indoor measurements may have to be modified for outdoor measurements. Some authors applied special treatments of SSNTDs (pre-etching, correction for ageing and annealing, proper choice of sheet of detectors) in order to increase the sensitivity and reduce the background of detectors.

Outdoor radon surveys were conducted at a much lower scale compared to indoor radon surveys. Only in a few papers, details regarding the design of the survey and quality assurance/quality control were given. Only a few maps of outdoor radon concentrations were produced. These maps could be potentially used as an input parameter for the estimation of indoor radon concentrations. However, maps were produced based on much smaller number of locations over larger grid cells, compared to indoor radon surveys and it is not clear if the values could be considered representative.

A few studies calculated correlation factors between indoor and outdoor radon concentrations. A wide range of obtained correlation factors indicates that there are additional parameters influencing measured quantities and outdoor radon concentration solely is not enough for indoor radon concentration prediction, but should be used simultaneously with other influencing factors.

Considering all the listed conclusions obtained in the overview of the literature regarding outdoor radon measurements campaigns, there is a strong need for harmonization of applied methods and international recommendations and protocols regarding outdoor radon measurements should be set up. This could be quite a challenge, since there are more goals of outdoor radon surveys compared to indoor radon ones, as outdoor radon concentrations are used for more applications than indoor radon concentration.

## Figures and Tables

**Figure 1 ijerph-19-00662-f001:**
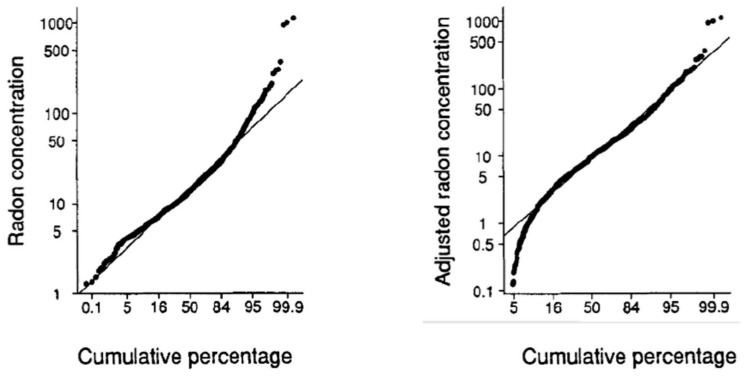
Uncorrected (**left figure**) and corrected (**right figure**) cumulative distribution of log of indoor radon concentration. Figure is taken from [55,56].

**Figure 2 ijerph-19-00662-f002:**
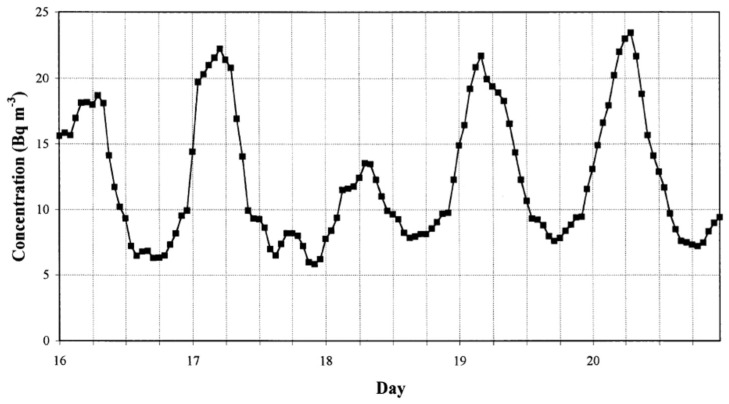
Diurnal variation of outdoor radon concentration from midnight 16 midnight 21 of August 2000 [66]. Sampling time was 1 h.

**Figure 3 ijerph-19-00662-f003:**
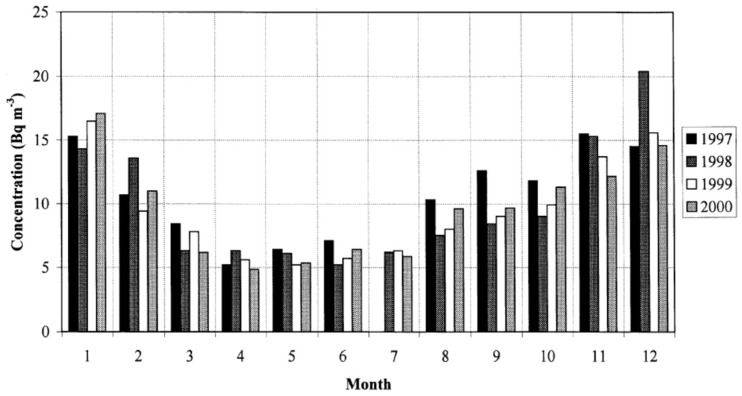
Monthly and year toyear variation of outdoor radon in Milan in period from 1997 to 2000. Figure is taken from Sesana et al. [66].

**Table 1 ijerph-19-00662-t001:** Properties of outdoor radon surveys worldwide.

Country	Type of Survey	No. of Locations	Measurement Protocol		
			Detector	Height [m]	Sampling Time
USA, Missouri [94]	National (in federal state)	82	CR-39	1 m	1 year
USA, Iowa [83]	National (in federal state)	111	CR-39	1.5 m	1 year
Minnesota [83]	Regional	64	CR-39	1.5 m	1 year
Turkey [77]	Regional	47 and 30	CR-39	N/A	1 year (1 det. in 3 months)
Slovenia [45]	National	60	CR-39	1.5	1 year (3 intervals)
China [79]	(9 cities)	101	LR115	1–1.5	A few times 2 months
China [89]	Nationwide (33 cities)	165	CR-39	1–3	1 year (3 intervals)
Serbia [46]	Regional	56	CR-39	1	1 year (4 intervals)
England [95]	Regional	69	CR-39	1–2	4 times 3 months
Norway [82]	NORM, 5 cites	82	Raduet (Rn/Tn)	1	Winter and summer
Lebanon [81]	Nationwide	24	EPERM	1	3 seasons (78 to 186 days)
Ireland [92]	National	18	CR-39	N/A	1 year
Japan [78]	National	696	Makrofol	1.5–2	4 times 3 months
Germany [85]	National	173	Makrofol	1.5	1
Iceland [91]	Local	1	Liquid scint.	N/A	May–July
Malta [73]	Local	3	PIPS	1	8 to 28 days
Cyprus [97]	Nationwide	11	Alpha guard	1	A few hours per meas.
East Asia [96]	Nationwide	20	CR-39	N/A	3 months
Syria [102]	Regional	36	Scintillator (Lucas cell)	N/A	10 min
Montenegro [93]	National	Theoretical			
	Local	1	Radtrack2	1	2 times 6 months
Spain [101]	National	25	Scintillator (ZnS:Ag)	highly variable	10 min

**Table 2 ijerph-19-00662-t002:** A descriptive statistics (range of measured concentrations (Range), geometric mean (GM), arithmetic mean (AM) and standard deviation (GSD)) of outdoor radon survey.

Country	No. of. Locations	Descriptive Statistics				Map
		Season	Range [Bq m^−3^]	GM (AM) [Bq m^−3^]	GSD [Bq m^−3^]	
USA, Missouri [94]	82	Annual	11–110	25	1.5	Yes
USA, Iowa [83]	111	Annual	7–55	29	1.4	Yes
Minesota [83]	64	Annual	4–55	19	1.8	Yes
Turkey [77]	47 30	Winter Summer	19–63.5 7–28	(34.10) (15.34)		
Slovenia [45]	60	Annual	3.7–41.0	11.8		Yes
China [79]	101	Annual	3.6–23.9	(9.3)		
China [89]	165	Annual	3–50	13.2 (14)		No
Serbia [46]	56	Annual	<244	49 (57)	1.8	No
England [95]	69	Annual		6	2	
Norway [82]	82	Winter Summer	4–13 8–210	(5–13) (29–82)		No
Lebanon [81]	24	Summer Autumn Winter	3.2–47.6 1.0–57.0 0.2–66.3	(19.7) (16.1) (13.4)		No.
Ireland [92]	18	Annual	4.2–7.7	(5.6)		No
Japan [78]	696	Annual	1.8–35.3	5.9 (6.1)		No
Germany [85]	173	Annual	3–31	9	1	Yes
Iceland [91]	1	May-july		1.6		No
Malta [73]	3	Summer	0.8–3.6			No
Cyprus [97]	12	August	2–134	9 (11)		No
East Asia [96]	20	3 months	5.3–17.0	(10.7)		No
Syria [102]	36	10 min.	5–66	21 (25)		No
Montenegro [93]	Theor.	Annual	6–11			
	1	Annual	13 ± 4	(13)		No
Spain [101]	25	Annual	1.2–15.8	(5.2)		No

**Table 3 ijerph-19-00662-t003:** Correlation of outdoor radon concentrations with other radon quantities.

Country	No. of. Locations	Correlation of Outdoor Rn with				
		Indoor Air	Soil Gas	Exhalation Rate	Gamma Dose Rate	^226^Ra in Soil
USA, Missouri [94]	82	0.11				
USA, Iowa [83]	111	0.2				
Minnesota [83]	64	0.7 (for 6 counties)				
Slovak Republic [123]	1	0.93				
India ^1^ [75]	1 (90 measurements)	0.9				
Turkey [77]	77	Weak				Weak
Slovenia [45]	60	Good	0.59			
China [79]	101		0.88			
Norway [82]	82				0.64	
Lebanon ^2^ [81]	24	0.10				
Japan [78]	696				0.62	
East Asia [96]	20			0.79		Weak
Syria ^2^ [102]	36	0.46	0.16	0.12		0.33
India [74]	1				−0.013	

^1^ correlation between indoor and outdoor EECRn. ^2^ correlations not given in the surveys, but calculated by authors of this review from available data in Lebanon and Syria [81,102].

## Data Availability

This is a review article. All data were gathered from the available literature.

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
