# Peer review of "Outdoor Radon as a Tool to Estimate Radon Priority Areas—A Literature Overview"

_ijerph, 2022, doi:10.3390/ijerph19020662_

Round 1
Reviewer 1 Report
If the article is to be of any benefit to the professional public, I suggest its substantial overhaul.
It is not clear from the paper why and how the measurements of radon in the outdoor atmosphere could help determine high values of indoor RAC. The most important source of radon in houses is the subsoil, and only in the third place is radon in the outdoor atmosphere, even more so because indoor radon in localities with a high concentration of radon in the outdoor atmosphere cannot fall below this value. However, even in this case, the source of radon in the outdoor atmosphere is the soil.
The overview is written rather chaotically and does not leave a coherent impression, it keeps jumping between indoor and outdoor atmosphere, some information is mentioned somewhere and a few lines later it is described again. The reader gets lost in the text.
The reasons for the seasonal variation of radon in the outdoor atmosphere are not described in detail, which is something a review study certainly requires.
Lines 153-170 - this chapter lacks some very important insights on the effects of mixing of air masses, atmospheric stability, and the influence of orography on radon in the outdoor atmosphere.
Chapter 3.1 in particular lacks references to peer-reviewed papers on measurements and analyses of diurnal radon cycles in the outdoor atmosphere published in the last 5 years. Many of these would have been beneficial for this review study.
Chapter 3.2 lacks an overview of a larger number of papers dealing with seasonal variations in both the indoor and outdoor atmosphere.
Lines 221-222
‘The data indicate that indoor and outdoor 221 values are negatively correlated with Pearson correlation factor of -0.18 [63].’ - this information is not universally valid! There are papers showing a positive correlation between indoor and outdoor atmosphere at the level of R=80-90. It is necessary to extend this section to include such results.
In Chapter 3.3, results from a 19-year continuous measurement in the outdoor atmosphere, carried out from 1991-2009 in one European country, could be mentioned.
Chapter 3.5 - RAC in the outdoor atmosphere also depends on the orography of the country - this information should also be mentioned, with reference to published papers.
Chapter 4 - I find it inappropriate to write how many measurements have been made with which detectors when the review study does not include all the papers published so far.
Of the whole chapter, lines 395-401 are the most beneficial; the other cited publications would also deserve such an analysis in the review study.
Measurements in the outdoor atmosphere are clearly needed; such measurements pose a challenge to the sensitivity of the detectors used. However, the use of outdoor measurements for radon priority areas estimation does not strike me as the most appropriate. More emphasis should be focused on other, more appropriate uses of radon in the outdoor atmosphere.
As an idea, the paper is not bad, but it needs to be substantially revised and include the most recent published papers on radon measurements and analysis in the outdoor atmosphere. I also recommend paying more attention to radon in the outdoor atmosphere and avoid jumping chaotically to radon in the indoor atmosphere or radon in the soil or radon exhalation from the soil.
Author Response
Authors's reply is reported in the attached document.

Reviewer 2 Report
This paper addressed the extensively surveyed review of outdoor radon variation. The study provides an important contribution to studies of atmospheric radon concentration. I recommend that the paper is accepted for publication after minor revision.
1) I think it is better to describe the vertical axis of the figure in detail.
Fig.1 Indoor radon concentration; Adjusted indoor radon concentration
Fig.2 Outdoor radon concentration
Fig.3 Outdoor radon concentration
2) L197 The subject of the sentence following "causing therefore" is difficult to understand.
The underlined part shows the proposed change.
This 24 h cycle is more pronounced in the summer during the cloudless days with light winds and large diurnal temperature range, compared to cloudy weather with a lower temperature gradient during the day and the night, causing therefore small or no variations of outdoor radon concentrations [57,61].
the night. Therefore, the latter had small or no variations of outdoor radon concentrations
3) L213 Please provide references for the following sentences.
Similarly, seasonal variation of radon behaviour can be observed. Diurnal variations in summer are more pronounced than during the winter days, as there are smaller temperature variations in winter.
4) I recommend the following papers.
4-1) Kumar’s studies describe the results of "radon strongly correlates with temperature, relative humidity, and a weak correlation with air pressure, ambient gamma dose during fair weather days"
Kumar, K. C.; Kamsali, N. One year of 222Rn concentration at a typical rural site in South India. Radiation Protection and Environment 2021, 44, 73–78.
4-2) UNSCEAR 2000 ANNEX B P99 Paragraph100desclibed following sentence.
“Data have shown that there were prominent increases in radon concentrations in outdoor air and in groundwater just before the large earthquake at Kobe, Japan, in 1995.”
This seismic radon variation in outdoor air was described the following paper. Kawada, Y. et al. Nonlinear Processes in Geophysics 2007, 14(2), 123-130.
I think the most famous of this seismic radon variation in groundwater is the following paper taken up in UNSCEAR 2000. Igarashi et al., Science, 269, 60-61.
5) Reference
5-1) I think the Abbreviated Journal Name needs a period.
Geosic Model Dev →Geosic. Model Dev.
5-2) No superscript or subscript applied.
222Rn→222Rn CO2→CO2
5-3) The way to indicate the issue number is not unified.
5-4) L749 last accessed →archived on
5-5) L806 S649-S655 → 649–655
Author Response

(The authors gave the same response as above.)

Round 2
Reviewer 1 Report
I have no comment.